# An Overview of Herbal-Based Antidiabetic Drug Delivery Systems: Focus on Lipid- and Inorganic-Based Nanoformulations

**DOI:** 10.3390/pharmaceutics14102135

**Published:** 2022-10-08

**Authors:** Espoir K. Kambale, Joëlle Quetin-Leclercq, Patrick B. Memvanga, Ana Beloqui

**Affiliations:** 1Advanced Drug Delivery and Biomaterials Group, Louvain Drug Research Institute, UCLouvain, Université Catholique de Louvain, Avenue Mounier 73, B1.73.12, 1200 Brussels, Belgium; 2Laboratory of Pharmaceutics and Phytopharmaceutical Drug Development, Faculty of Pharmaceutical Sciences, University of Kinshasa, B.P. 212, Kinshasa 012, Democratic Republic of the Congo; 3Pharmacognosy Research Group, Louvain Drug Research Institute, UCLouvain, Université Catholique de Louvain, Avenue Mounier 72, B1.72.03, 1200 Brussels, Belgium; 4Centre de Recherche et d’Innovation Technologique en Environnement et en Sciences de la Santé (CRITESS), University of Kinshasa, B.P. 212, Kinshasa 012, Democratic Republic of the Congo; 5Walloon Excellence in Life Science and Biotechnology (WELBIO), Avenue Pasteur 6, 1300 Wavre, Belgium

**Keywords:** oral delivery, antidiabetic phytocompounds, lipid-based nanoparticles, inorganic nanoparticles

## Abstract

Diabetes is a metabolic pathology with chronic high blood glucose levels that occurs when the pancreas does not produce enough insulin or the body does not properly use the insulin it produces. Diabetes management is a puzzle and focuses on a healthy lifestyle, physical exercise, and medication. Thus far, the condition remains incurable; management just helps to control it. Its medical treatment is expensive and is to be followed for the long term, which is why people, especially from low-income countries, resort to herbal medicines. However, many active compounds isolated from plants (phytocompounds) are poorly bioavailable due to their low solubility, low permeability, or rapid elimination. To overcome these impediments and to alleviate the cost burden on disadvantaged populations, plant nanomedicines are being studied. Nanoparticulate formulations containing antidiabetic plant extracts or phytocompounds have shown promising results. We herein aimed to provide an overview of the use of lipid- and inorganic-based nanoparticulate delivery systems with plant extracts or phytocompounds for the treatment of diabetes while highlighting their advantages and limitations for clinical application. The findings from the reviewed works showed that these nanoparticulate formulations resulted in high antidiabetic activity at low doses compared to the corresponding plant extracts or phytocompounds alone. Moreover, it was shown that nanoparticulate systems address the poor bioavailability of herbal medicines, but the lack of enough preclinical and clinical pharmacokinetic and/or pharmacodynamic trials still delays their use in diabetic patients.

## 1. Introduction

Herbal-based antidiabetic drug delivery systems refer to technologies for delivering herbal drugs to achieve a therapeutic effect against diabetes. In fact, diabetes is a chronic and metabolic disease that occurs when the pancreas does not produce enough insulin or the body does not properly use the insulin it produces [1]. It is characterized by a chronic high glucose level in the blood [2]. Today, diabetes remains incurable [3]. The management helps only to control the condition, balancing the benefits against the side effects of medication [4,5].

Since there is no cure for diabetes, its management is very expensive, and nowadays, about 90% of people with undiagnosed diabetes live in low- and middle-income countries [1]. They use herbal medicines to improve their quality of life because of their benefits (easy accessibility, safety, cost-effectiveness, and marvelous antidiabetic potential) [6,7]. To encourage this practice, several herbal medicines are described in the World Health Organization (WHO) monographs of selected plants [8]. Thus, in recent years, a considerable number of studies have been conducted on herbal medicines to maximize their antidiabetic capacity. Ongoing research focuses, in particular, on oral drug delivery. On the one hand, some aim to increase the antidiabetic activity or bioavailability of phytocompounds by developing formulations with particle sizes reduced to nanometers. On the other hand, due to the high reducing potential of plant compounds, plant extracts have become useful in the biological synthesis (green synthesis) of antidiabetic metal nanoparticles by acting as reducing, capping, and stabilizing agents but also by contributing to their biological activity [9,10]. To understand the contribution of these preparations to improving the antidiabetic efficiency of herbal medicines in the treatment of diabetes, it is important to review the available literature on their use and the increased therapeutic value they have provided. Herein, we aim to provide an overview of the use of lipid- and inorganic-based (metal or metal oxide) nanoparticulate delivery systems with plant extracts or phytocompounds for the treatment of diabetes while highlighting their advantages and limitations for clinical application.

## 2. Diabetes Importance and Treatment

Due to chronic high blood glucose, diabetes is accompanied by an increased formation of free radicals and a decrease in antioxidant potential, which in the long term leads to a number of complications, including retinopathy, cardiovascular disease, diabetic foot, neuropathy, and nephropathy [11,12]. It is a serious health problem whose incidence rate grows continuously and causes many deaths worldwide. The International Diabetes Federation has indicated that 537 million adults (20–79 years) lived with diabetes in 2021, and these figures are expected to rise to 783 million by 2045. This federation also reported 6.7 million deaths due to diabetes complications in 2021 [1]. Diabetes can be subdivided into different types, among which the most important are type 1, type 2, and gestational [13]. Type 1 diabetes mellitus, also called juvenile-onset diabetes, is a condition in which there is no or little production of insulin by pancreatic β-cells. It is caused by genetic or infectious factors leading to autoimmune damage to β-cells. It is the main type of diabetes in childhood or adolescence but can appear at all ages. People living with type 1 diabetes account for approximately 5% to less than 10% of all diabetes cases. Type 2 or noninsulin-dependent diabetes is characterized by defective use of insulin by cells. It is caused by inheritance factors, overweight, and lifestyle factors and accounts for up to approximately 90% of diabetic people. Gestational diabetes occurs during pregnancy, is caused by hormonal changes, and accounts for less than 5% of cases [1,14,15].

Symptoms of diabetes can include increased thirst, frequent urination, extreme hunger, fatigue, unexplained weight loss, and vision disturbance [16,17]. These symptoms appear suddenly in type 1 but slowly in type 2 diabetes. The current treatment for type 1 diabetic patients is mainly based on exogenous insulin supply [3,18], and blood glucose monitoring, physical activity, and healthy diet practices are recommended for these patients [1]. Alongside insulin, pramlintide (an injectable amylin analog) can be used in adults with type 1 diabetes to decrease the pancreatic secretion of glucagon, a hormone that acts mainly on the liver by causing glycogenolysis, by α2-cells of the islets of Langerhans [19,20]. For type 2 diabetic patients, among the strategies used, regular physical exercise, proper low-carbohydrate diets, and subsequent medication therapy are recommended [1,21]. Medicines used for type 2 diabetes include different families of oral hypoglycemic agents (whose representatives are in parentheses), such as biguanides in first-line therapy (metformin), sulfonylureas (glibenclamide, glimepiride, gliclazide glipizide, glyburide), meglinides (repaglinide and nateglinide), thiazolidinedione (TZD) (rosiglitazone and pioglitazone), dipeptidyl peptidase 4 (DPP-4) inhibitors (sitagliptin, saxagliptin, vidagliptin, linagliptin, and alogliptin), sodium-glucose cotransporter (SGLT2) inhibitors (canagliflozin, dapagliflozin, and empagliflozin), α-glucosidase inhibitors (acarbose), glucagon-like peptide-1 analog (semaglutide), and injectables such as insulin and glucagon-like peptide-1 agonists (exenatide, liraglutide) [19,22]. The management of gestational diabetes is based on lifestyle and dietary measures, with recourse to insulin therapy in case of failure [23]. In addition to these conventional treatments are plant-based alternative medicines. Herbal therapy, due to the presence of phytocompounds such as flavonoids or other polyphenols, alkaloids, saponins, terpenoids, polysaccharides, quinones, etc. [24], which exert hypoglycemic and antioxidant effects [25], is practiced. Numerous Food and Drug Administration (FDA)-approved drugs are derived either directly or indirectly from plants. As examples of these drugs, we can cite (i) biguanides (metformin), developed from galegine, an alkaloid isolated from *Galega officinalis* L. [26,27]; (ii) canagliflozin, dapagliflozin, and empagliflozin, which are synthetic compounds derived from phlorizin, a flavonoid from apple tree bark [28,29]; (iii) morphine, an alkaloid extracted from opium resin [30]; and (iv) diosmin, a flavonoid from a lemon tree and citrus zest [31], respectively.

## 3. Herbal-Based Antidiabetic Medicines and Overview of The Antidiabetic Activity of Some Phytocompounds

The use of herbal medicines comes from traditional medicine, which is a set of practices, knowledge, and beliefs based on explicable or unexplainable theories that are specific to each culture and applied for the prevention, management, and diagnosis of physical and mental ailments [32]. These practices are transmitted from generation to generation either spiritually by ancestors or acquired by past experience, observations, and learning [33]. For medical care in traditional medicine, plants, minerals, animal parts, incantations, and other practices are used, and among them, medicinal plants constitute the mainstay of disease treatment [34].

Phytocompounds can come from many different parts of plants, including leaves, fruit, seeds, flowers, exudate, bark, and roots used in either raw-form dietary supplements or extracts (crude or purified) [35]. Within one plant, different parts can contain different concentrations of active phytochemicals. For example, in *Ficus capensis*, more flavonoids are found in leaves than in roots, and more tannins are found in roots than in leaves [36]. It is also observed that different parts of a plant can be used against different diseases. *Mangifera indica* is a well-known example whose bark is used to treat, among other conditions, diabetes, while the leaves cure diseases of the lungs, coughs, and asthma [37].

Plants have long been used and occupy important places in human life as food, medicine, places of habitation, and animal feed [38]. Medicinal plants are commonly used in all countries, and WHO reports indicate that in developing countries, nearly 80% of the population uses plants for primary care. This is the case for people in Africa and Asia [39]. In rural areas, availability of medicinal plants and few or no known side effects make herbal preparations the primary treatment for diabetes [40]. Traditional medicine somewhat overcomes the cost burden of conventional drugs.

Many studies have demonstrated the beneficial effects of medicines made from plants, also called phytomedicines, on type 1 and type 2 diabetes [41,42]. Plant-derived metabolites exert their effects via various mechanisms (Figure 1), including prevention of glucose absorption from the intestine due to the inhibition of intestinal α glucosidase and pancreatic α amylase, functional β cell restoration, insulin secretion, and expression improvement, decrease in insulin resistance, glucose use promotion, or the prevention of glucose production from liver cells, regulation of carbohydrate and lipid metabolism, inhibition of the dipeptylpeptidase-4 enzyme, and increasing antioxidant enzyme activity [11,43,44,45].

Herbal medicines, when associated with a healthy lifestyle, are used in the management of diabetes and generally improve patient conditions [41]. Their numerous antidiabetic properties have attracted much attention from researchers and are the subject of recent publications. In this regard, Salehi et al. listed several antidiabetic plants commonly used in different regions around the world and described the antidiabetic potential of their isolated phytocompounds [24]. Among these commonly used plant-extracted compounds, some have unfavorable bioavailability. Among them, some have been incorporated into nanoparticulate formulations to study the improvements in their bioavailability and antidiabetic activity. They include phenolic compounds (quercetin, curcumin, naringenin, resveratrol, ferulic acid, myricetin, liquiritin, baicalin), nitrogenous compounds (berberine, betanin), terpenic compounds (glycyrrhizin, gymnemic acids, thymoquinone, lutein, δ-oryzanol), and miscellaneous compounds (bitter gourd seed oil, polypeptide-k, sage oil).

### 3.1. Phenolic Compounds

#### 3.1.1. Quercetin

Quercetin is a polyphenolic compound found in various plants, including apples, onions, seeds of tomatoes, berries, citrus fruits, buckwheat tea, and fennel [46]. Studies have reported that it has high antioxidant and anti-inflammatory activities [47] and is involved in many antidiabetic actions, such as insulin secretion and sensitization, glucose level improvement, and inhibition of intestinal glucose absorption [48]. Quercetin stimulates glucose transporter 4 (GLUT4), the main facilitative mediator of glucose uptake in skeletal muscles, adipose tissues, and other peripheral tissues, by activating adenosine monophosphate and avoiding free radical formation by inhibiting lipid peroxidation; it also acts as an anti-allergic, anticancer, anti-ulcer, cardiovascular protector, and preventive agent against retinopathy [47]. It can be used to prevent changes in blood glucose and body weight [49]. In a study in rats with type 2 diabetes, a single oral dose of quercetin (400 mg) inhibited α-glucosidase activity and showed a decrease in postprandial hyperglycemia [50]. A recent meta-analysis of patients with metabolic disorders concluded that quercetin can significantly reduce plasma glucose levels when the dose is greater than 500 mg/day after more than 8 weeks. Thus, the dose and duration of treatment are important factors for the effectiveness of this treatment [51]. Up to 2000 mg/day, quercetin has been shown to be safely tolerated and is approved for human use [52,53]. However, due to its poor solubility, its absorption is low in the gastrointestinal tract, and only approximately 2% to 6.7% is absorbed after oral administration [54,55]. Additionally, its plasmatic peak is reached in 0.7–7.0 h [47].

#### 3.1.2. Curcumin

Curcumin is the major active curcuminoid and a natural polyphenol of turmeric. It comes from the roots of *Curcuma longa*, which has long been used in Ayurvedic and traditional Chinese medicine and has received a great deal of attention for its medicinal virtues, including antioxidant, anti-inflammatory, and antihyperglycemic properties [56,57]. Several studies have shown that curcumin improves insulin sensitivity, decreases hyperglycemia, improves insulin levels, reduces proinflammatory mediators such as interleukin (IL-6, IL-1β) and tumor necrosis factor-α (TNF-α) in diabetic patients, and increases their global antioxidant power [58,59]. It activates adenosine monophosphate-activated protein kinase (AMPK) enzyme and inhibits glucose 6-phosphatase (G6Pase) activities [60]. In a clinical study in which curcumin was administered orally for 3 months at a dose of 300 mg/day to overweight and obese type 2 diabetic patients, Na et al. observed decreases in body mass index (BMI), fasting blood glucose, glycosylated hemoglobin, insulin resistance index (HOMA-IR), and free fatty acids. Compared to the baseline, the observed reduction in glucose levels was 18%, and that for glycosylated hemoglobin was 11% [61]. Curcumin is recognized as safe by the Food and Drug Administration (FDA) [62] and is safely tolerates up to 8 g per day [63]. Although curcumin has a safe recognition and various physiological effects, it has a low oral bioavailability of approximately 1% due to low digestive absorption and therefore undergoes excretion [64]. This drawback limits its therapeutic efficacy [57].

#### 3.1.3. Naringenin

Naringenin is a polyphenol (more precisely a flavonoid) found mainly in citrus fruits, with much higher amounts found in grapefruits and oranges [65]. In vitro and in vivo studies have reported that naringenin has several pharmacological activities, including antioxidant and hepatoprotective activities, due to free radical scavenging, boosting endogenous antioxidant activity and metal ion chelation [66]. It was also reported to be anti-inflammatory by inhibiting cytokines and inflammatory transcription factors (TN-α, IL-6) [67] and antidiabetic by increasing insulin sensitivity, GLUT4 activity, and glucose uptake by activating AMPK [65]. Sharma et al. investigated different doses of naringenin: 25, 50, and 100 mg/kg per day were administered orally to type 2 diabetic rats for 28 days. There were significant decreases in insulin resistance, hyperinsulinemia, hyperglycemia, dyslipidemia, TNF-α, and IL-6 and concomitant increases in adiponectin and β-cell function. These activities were found to be dose dependent [68]. However, naringenin is soluble in organic solvents, such as alcohol [69], and suffers from poor solubility in water and low absorption in the intestine, leading to poor bioavailability (5.81 ± 0.81%) [70] and quick elimination after oral administration [71]. Studies in Beagle dogs for 6 months at a dose of 500 mg/kg body weight per day orally showed that it was safe [72]. In adult humans, a single oral dose of 900 mg was also shown to be safe [73].

#### 3.1.4. Resveratrol

Resveratrol is a natural polyphenolic compound that belongs to the stilbenoid class and is extracted from many plants, including grapevines, berries, peanuts, giant knotweed, and the *Polygonum cuspidatum* plant rhizome [74]. Plants containing resveratrol have been used for more than 2000 years in traditional medicine [75]. Resveratrol has been reported to have strong antioxidant and antidiabetic activities. It inhibits α-amylase and α-glucosidase activity, improves insulin sensitivity, and increases glucose uptake and storage [76]. Through its antioxidant properties, increasing antioxidant enzyme activities prevents pancreatic β-cell apoptosis and dysfunction [77]. In a review of clinical studies on the efficacy of resveratrol on hyperglycemic status in type 2 diabetic patients, Movahed et al. showed that an oral dose of resveratrol of 1 g per day for 45 days significantly decreased fasting blood glucose, insulin, and systolic blood pressure from 175.7 mg/dL, 10.2 µ international unit (IU)/mL, and 129 mmHg to 140.1 mg/dL, 5.4 µIU/mL, and 121.5 mmHg, respectively [78]. A safety study by Tani et al. showed that 5 g of resveratrol per day was well tolerated in humans [79]. Despite its strong activities, poor bioavailability (less than 1%) and poor light and heat stability are major drawbacks for clinical usage [77,80].

#### 3.1.5. Ferulic Acid

Ferulic acid is a phenolic phytochemical found in many commonly consumed foods, such as cereals, whole-grain foods, banana, coffee, orange juice, eggplant, bamboo shoots, beetroot, citrus, cabbage, broccoli, and spinach [81], and is known for its high antioxidant and antidiabetic properties [81,82]. Ferulic acid has been reported to inhibit α glucosidase and stimulate insulin secretion and cell sensitivity to insulin [83,84,85]. Ferulic acid is rapidly absorbed (approximately 90%) after oral ingestion and rapidly reaches its plasma peak concentration in 30 min [86]. In a clinical study reported by Bumrungpert et al. in hyperlipidemic subjects, ferulic acid was administered orally 1000 mg per day for six weeks. Ferulic acid demonstrated statistically significant decreases in total cholesterol by 8.1% and triglycerides by 12.1%, an increase in HDL cholesterol by 4.3%, a decrease in malonylaldehyde by 24.5%, and a decrease in TNF-α by 13.1% compared to the placebo [87]. These data demonstrated that ferulic acid can improve the condition of hyperlipidemic diabetic patients. Ferulic acid was shown to possess very low toxicity with LD50 of 2445 mg per kg and 2113 mg per kg in male and female rats, respectively [88]. It is approved in some counties, including Japan, as an antioxidant and is added to food products [81]. However, while possessing beneficial antihyperlipidemic, antioxidant, and antidiabetic properties, ferulic acid undergoes an important first-stage passage metabolism, and its metabolites are rapidly excreted in urine [89]. Only 9% to 20% of unmetabolized ferulic acid is found in the plasma, and this low, inadequate plasma concentration elicits weak biological responses [90].

#### 3.1.6. Myricetin

Myricetin is a natural polyphenolic compound found mainly in fruits, berries, vegetables, teas, wine, and plants from the family Myricaceae (bark of *Myrica nagi*) and other families, such as Anacardiaceae, Pinaceae, Primulaceae, and Polygonaceae [91]. It is used in beverages as an additive, and the Food and Agriculture Organization (FAO) and the U.S. Flavor and Extract Manufacturer Association (FEMA) have recognized it as safe [92]. In vitro and in vivo studies reported its antidiabetic activities. In an in vitro study reported by Tadera et al., myricetin considerably inhibited α-glucosidase and α-amylase [93]. In diabetic rats, myricetin has been shown to improve renal functions and the antioxidant activities of glutathione peroxidase and xanthine oxidase enzymes [94]. Another study by Ong and Khoo reported that in a diabetic mouse model, treatment with 3 mg every 12 h orally with myricetin for 2 days resulted in a 50% reduction in blood glucose to normal in comparison with the control [95]. However, after oral administration, myricetin is poorly soluble [96]. Dang et al., in a study on rats, found low bioavailabilities of 9.62% and 9.74% at oral doses of 50 and 100 mg/kg, respectively [97].

#### 3.1.7. Liquiritin

Liquiritin is a phytocompound extracted from Glycyrrhiza species [98,99], plants with antidiabetic properties [100,101]. Data found on liquiritin showed that oral administration of 10 or 20 mg in mice decreased insulin resistance, improved lipid metabolism, reduced the levels of inflammatory factors, including IL-β, IL-1β, IL-18, TNF-α, and IL-6, increased antioxidant enzyme levels of superoxide dismutases 1 and 2 (SOD1 and SOD2), and decreased myocardial fibrosis by inhibiting the nuclear factor-kappa B (NF-κB) and mitogen-activated protein kinase (MAPK) signaling pathways [102]. However, pharmacokinetics studies by Han et al. showed that it is poorly bioavailable. Indeed, after 1 g/kg oral administration of *Glycyrrhizae radix* extract in rats, the maximum plasma concentration of liquiritin was found to be very low (39.5 ± 7.8 ng·h/mL) [103]. In a safety study of liquiritin at 300 mg/kg for a week, intraperitoneally injected mice showed no sign of toxicity [104].

#### 3.1.8. Baicalin

Baicalin is a flavonoid extracted from *Scutellaria baicalensis*, a medicinal plant used to treat various pathologies, including inflammatory diseases [105]. Proinflammatory response suppression is attributed to its flavonoids, of which baicalin is the major member [106]. It has been reported to possess antidiabetic, antioxidant, and anti-inflammatory effects by activating insulin receptor substrate-1 (IRS-1) and GLUT-4, and involving AMPK signaling cascade [107], inhibiting α-glucosidase [108] and the production of TNF-α, IL-6, and other substances induced by lipopolysaccharide (LPS) [105,106]. In an examination of the effect on mice by Wang et al., baicalin 4 mg/kg per day was administered by the intragastric route to streptozocin-induced diabetic pregnant mice. In comparison to the diabetic controls, the glycemia was significantly lowered by treatment with baicalin after 3 weeks of treatment (*p* < 0.01) [109]. In a study by Li et al., oral baicalin at a dose up to 600 mg per day was well-tolerated safe in healthy subjects [110]. Despite its activities, baicalin has shown poor water solubility, poor absorption, and high biliary excretion in in vivo studies, leading to poor bioavailability (approximately 3%) [111,112].

### 3.2. Nitrogenous Compounds

#### 3.2.1. Berberine

Berberine is an alkaloid belonging to the isoquinone group [113]. It is found in various plants, including *Berberis vulgaris*, *Coptis deltoidei, Coptis teetoides, Berberis amurense*, and *Phellodendron amurense*, and has been used for the treatment of various diseases, including diabetes [114]. Studies have shown that berberine reduces fasting blood glucose, postprandial blood glucose, glycated hemoglobin, plasma triglycerides, and total cholesterol [115]. Berberine exerts its hypoglycemic activity in different ways. It activates the AMPK pathway, increases the sensitivity of receptors to insulin, decreases insulin resistance, regulates glucokinase, increases glucose transporter activity, improves glucose uptake, inhibits gluconeogenesis, and promotes glycolysis [114,115,116,117,118]. Shende et al. reported a study of 30 newly diagnosed type 2 diabetic patients receiving either metformin 500 mg every 12 h or berberine 500 mg every 12 h orally for 12 weeks. The parameters studied included fasting blood glucose (FBG), postprandial blood glucose (PPBG), and glycated hemoglobin (HBA1c). Berberine significantly reduced FBG levels from 156.2 ± 11.3 mg/dL to 136.4 ± 9.7 mg/dL, PPBG from 316.8 ± 25.6 mg/dL to 197.4 ± 14.7 m/dL, and HBA1c from 7.6 ± 0.5% to 7.1 ± 0.5% and significantly improved the lipid profile. The results were similar to those obtained with metformin [119]. Berberine is considered safe and is approved by the FDA as an antihyperglycemic agent [120]. Despite its remarkable antidiabetic effects, berberine is poorly absorbed in the gastrointestinal tract, and the absorbed fraction (nearly half) suffers from the effect of the first hepatic passage [121]. This poor absorption leads to low bioavailability and elimination of a considerable portion of berberine.

#### 3.2.2. Betanin

Betanin is a water-soluble red-violet pigment, and major betalain is mainly found in beetroot (*Beta vulgaris* L) [122] and other plants, such as amaranth (Amaranthaceae), cactus pear (*Opuntia spp*.), djulis (*Chenopodium formosanum*), pitahayas (*Hylocereus undatus*), and pitayas (*Stenocereu spp*.) [123]. Betanin is approved by the European Food Safety Authority (EFSA) and the U.S. Food and Drug Administration (FDA) as a natural colorant and possesses antioxidant and antidiabetic activities [124,125]. It activates the AMPK pathway and reduces nuclear factor-κB mRNA expression [126]. Due to betanin, beetroot has been reported to possess antidiabetic, antioxidative, and anti-inflammatory activities and is traditionally used to treat diabetes [125,126]. Administration of 2 g/kg *Beta vulgaris* extract orally every day to diabetic rats resulted in a 50% decrease in blood glucose levels in a study by Yanardag et al. [127]. Additionally, in another assessment, a 40% blood glucose reduction without resulting in weight loss or hepatic impairment was observed in diabetic rats after administration of the same dose [128]. However, its bioavailability after oral administration was found to be very low in a study in humans, suggesting that it is primarily lost during the digestive process [129].

### 3.3. Terpenic Compounds

#### 3.3.1. Glycyrrhizin

Glycyrrhizin is a triterpenoid saponin extracted from the roots of various species of Glycyrrhiza (*Glycyrrhiza uralensis*, *Glycyrrhiza glabra*, or *Glycyrrhiza inflata*) that is more than 30 times sweeter in flavor than saccharose but does not increase glycemia [130]. Glycyrrhizin has been used for a long time in herbal medicine due to its multiple properties, including antioxidant [131] and antidiabetic properties [132] and is generally recognized as safe (GRAS) by the FDA [133]. It upregulates the peroxisome proliferator-activated receptor gamma (PPAR-γ) and GLUT4 and is reported to reduce the postprandial increase blood glucose [134,135]. Glycyrrhizin is quite soluble but poorly bioavailable after oral administration due to low and incomplete absorption [136]. Wang et al., in their investigations, found its oral bioavailability to be 4% and approximately 14.2% for glycyrrhizic acid, its active metabolite (resulting from hydrolysis) [137]. In another study by Baltina et al., 100 mg/kg glycyrrhizic acid administered orally to rats showed a 35.5% reduction in blood glucose levels after 2 h [138].

#### 3.3.2. Gymnemic Acids

Gymnemic acids are glycosides of triterpenes from *Gymnema sylvestre* leaf extracts that have been used worldwide for years as an herbal remedy mainly for their antidiabetic and hypolipidemic properties [139]. In vivo studies have shown that gymnemic acids increase insulin secretion by enhancing beta cells membrane insulin permeability [140], decrease blood glucose levels in patients with type 2 diabetes [141], and protect β cells against oxidative stress [142]. Some clinical trials using *Gymnema sylvestre* extracts have confirmed these activities. In a study reported by Khare et al., 2 g of aqueous extract administered three times daily to diabetic patients for 15 days reduced fasting blood glucose and two-hour orally induced hyperglycemia from 135.7 mg/dL to 110.7 mg/dL and from 152.7 to 151.1 mg/dL, respectively [143]. However, their poor water solubility, poor lipid solubility, and very poor oral bioavailability have been reported [144,145]. The limited clinical information available indicates that high doses of *Gymnema sylvestre* extracts are potentially toxic [144]. Therefore, all these characteristics limit their wide clinical use.

#### 3.3.3. Thymoquinone

Thymoquinone is a volatile oily compound extracted from the seeds of *Nigella sativa* that has been used for years in the medical and culinary fields due to its multiple beneficial effects [146]. Among them, one can cite antidiabetic, antioxidant, hypolipidemic, anti-inflammatory, and anticancer activities [147]. Thymoquinone modulates PPAR-γ and GLUT4 expression and inhibits the glycogen phosphorylase enzyme involved in glycogenolysis [148,149]. It has been reported to decrease blood glucose levels and oxidative stress in a rat model [150], decrease fasting insulin levels, improve glucose tolerance and insulin sensitivity, and reduce hepatic glucose production [151]. A clinical study on glycemia control with thymoquinone using *Nigella sativa* seeds as powder in capsules evaluated the effect on uncontrolled type 2 diabetic patients. The study showed that administration of 2 g of powder per day for 4, 8, and 12 weeks resulted in significant reductions in fasting blood glucose of 45, 62, and 56 mg/dL, respectively, from 195.95 mg/dl baseline and 1.52% for glycated hemoglobin at the end of the 12 weeks of treatment [152]. The studies by Najmi et al. also reported significant decreases in these glycemic parameters [153]. A clinical study by Thomas et al. in healthy adults reported that *Nigella sativa* oil containing 5% thymoquinone is safe at a dose 200 mg/kg per day for 90 days [154]. Nonetheless, thymoquinone is poorly soluble in water, highly lipophilic, weakly absorbed, quickly eliminated, poorly bioavailable, and unstable in light, high temperatures, and extreme pH [155,156]. Alkharfy et al. reported a slow absorption of thymoquinone and an absolute bioavailability of approximately 58%, with more than 99% bound to serum proteins, leading to its rapid elimination after oral administration [157].

#### 3.3.4. Lutein

Lutein is a carotenoid phytocompound found together with zeaxanthin at high quantities in parsley, spinach, kale, and egg yolk. It has high power to scavenge reactive oxygen, and due to its beneficial antioxidant effects, it is used in the management of diabetes complications such as diabetic retinopathy [158]. It is approved by the FDA as GRAS and has been reported to prevent oxidative damage to the retina [159,160] and to prevent the effects of high blood glucose on the immune system, reducing the engagement of nuclear factor-kappa B (NF-κB) in the immune response and cell death [161]. A study by Katyal et al. demonstrated that lutein effectively prevents the development and progression of diabetic nephropathy. A dose of 1.5 mg/kg per day for 4 weeks was administered orally and significantly lowered blood sugar and increased antioxidants [162]. Despite its important properties, lutein is poorly soluble in water and poorly bioavailable after oral administration, which leads to difficulties in achieving a biologically effective dose in food sources [163].

#### 3.3.5. γ-Oryzanol

γ-Oryzanol is a ferulic acid ester of a tetracyclic triterpene extracted from brown rice and is the major constituent of rice oil [164]. γ-Oryzanol has been reported to possess antidiabetic [165] antioxidant [166] activities and is considered safe by the FDA. The consumption of brown rice is known to prevent postprandial hyperglycemia. In vivo studies by Kozuka et al. reported that γ-oryzanol protects beta cells by reducing the transcriptional activity in the endoplasmic reticulum stress-responsive element, improves glycemia, and stimulates insulin secretion by beta cells [167]. γ-Oryzanol (3.2 mg/g body weight, oral administration) significantly decreased beta cell apoptosis and improved their functions. However, γ-oryzanol is poorly soluble and has low bioavailability [166,168].

### 3.4. Miscellaneous

#### 3.4.1. Bitter Gourd Seed Oil and Polypeptide-k

Both seed oil and polypeptide-k are extracted from *Momordica charantia* (bitter gourd), a plant with a long history of use as both food and medicine in several countries [169]. *Momordica charantia* has been used largely against diabetes, and fruit extracts activate adenosine monophosphate-activated protein kinase (AMPK), enabling glucose uptake [170]. Approximately 50% of seed oil is represented by α-eleostearic acid, which has antioxidant and antidiabetic activities [171,172]. Both seed oil and polypeptide-k possess important in vitro α-glucosidase and α-amylase inhibition activities and lower blood glucose concentrations by acting on peripheral tissues [169]. Moreover, polypeptide-k has shown insulin-like effects and is considered a phytoinsulin [173]. A study by Lok et al. on 18 healthy humans showed hypoglycemic activity due to 2 mg polypeptide-k-supplemented soft buns. Two hours after oral administration, a decrease of 0.9 mmol/L in blood glucose was observed [174]. However, bitter gourd seed oil and polypeptide-k are poorly soluble and have low absorption in the gastrointestinal tract and weak bioavailability. Thus, an adequate form is needed to achieve therapeutic effects [173,175]. In addition, despite their long use in traditional medicine, there are no safety studies to our knowledge.

#### 3.4.2. Sage Oil

Sage oil is the essential oil of *Salvia officinalis*, a plant used for years in many countries in the treatment of several diseases, including diabetes [176]. *Salvia officinalis* metabolites also include polyphenols and diterpenes, which may in part be responsible for plant antidiabetic activities [177,178]. Recently, in vivo studies by Elseweidy et al. showed that *Salvia officinalis* oil intake considerably lowered hyperglycemia and possessed antibacterial, antioxidant, and anti-inflammatory activities. In these studies, intraperitoneal administration of sage oil (0.042 mg/kg body weight daily) in zinc-deficient diabetic rats for 8 weeks decreased serum glucose to 94.17 mg/dL, while the serum glucose level of the zinc-deficient control group was 547 mg/dL [179]. A toxicological study of sage oil reported that more than 0.5 g/kg can lead to convulsion due to its action on the nervous system [180]. However, sage oil is slightly soluble and is composed mainly of thujone, camphor, 1,8-cineole, α-humulene, α-pinene, camphene, and bornyl acetate, which are poorly soluble [181].

### 3.5. Limitations of Current Antidiabetic Phytocompounds

As described above, studies of numerous plant secondary metabolites (sometimes used in purified form or in the form of poorly purified extracts) have shown their effects on managing diabetes. Their preventive or alleviative effects against associated complications have also been shown. For example, ophthalmic neuropathy can be preventable by lutein [158], cardiovascular disease and retinopathy by quercetin [47], and liver damage by naringenin [66]. These medicines are mostly administered by the oral route. However, their low solubility in water, low absorption, low permeability, low systemic availability, fast metabolization, instability in the gastrointestinal tract or in different environmental conditions, nonspecific distribution to organs, and high elimination still hinder their clinical use as therapeutic agents for diabetes [11,182,183]. Therefore, significant quantities must therefore be ingested to reach the therapeutic effect. Another limitation is the possible drug–drug interactions due to their multiple components (for example, in sage oil) and the lack of in-depth toxicity studies for long-term use.

Since many active compounds of plant origin have unfavorable bioavailabilities, different types of nanoparticle formulations (see next section) have been used to overcome this shortcoming.

## 4. Current Lipid- and Inorganic-Based (Metal or Metal Oxide) Nanoformulations for Antidiabetic Herbal Products

Nanoformulations, i.e., nanoparticlulate systems are colloidal dispersions of very small size ranging from 1 to 1000 nm, with a high surface area in relation to their volume, and whose morphology and properties depend on their components and preparation methods [184,185]. They have several applications, including their use as drug carriers. They are highly valued, as they impart new properties and characteristics not found in the bulk material itself [186]. They facilitate the penetration and crossing of biological barriers, thus improving the bioavailability of therapeutic agents [187]. Researchers are currently developing nanoparticle formulations containing antidiabetic plant products (Figure 2), with a view of going beyond the limits of their traditional use.

Nanoparticle-based formulations containing herbal medicines have shown a series of excellent benefits over their bulk counterparts, such as suitable bioavailability, biocompatibility, low toxicity, and targeting efficiency [44]. Examples of these formulations for diabetes treatment that are being developed and have been tested include lipid-based nanocarriers and inorganic nanocarriers described below.

### 4.1. Lipid-Based Nanoparticle Formulations Containing Antidiabetic Crude Plant Extracts or Phytocompounds

Lipid-based nanocarriers are designed with nanoparticulate formulations carrying a dissolved or suspended drug in lipidic excipients with or without surfactants added [188]. Their excipients are generally recognized as safe (GRAS) by the FDA [188,189]. Thanks to this recognition and the advantages offered by lipid-based nanocarriers, including reasonable cost, success to be loaded with both lipophilic and hydrophilic substances, remarkable encapsulation efficiency, controlled or programmed release of the drug, ease of production for almost all formulations, increased bioavailability of the drug, and suitability for various administration routes (oral, intravenous, intramuscular, pulmonary, etc.), lipid-based nanoparticulate systems are currently the most commonly prepared nanoparticulate systems [190,191,192]. Lipid-based nanoparticle formulations in which plant-derived antidiabetic drugs have been incorporated are mainly used as carriers to improve bioavailability; they can also be used to protect the plant-extracted drug by encapsulating it, masking unpleasant flavors, achieving controllable release and efficiently delivering the active compound to the target tissues [192]. Examples of the outcomes of in vivo evaluations of plant-extracted antidiabetic drugs loaded into lipid-based nanoparticulate carriers compared to unloaded carriers are summarized in Table 1. Lipid-based nanoparticulate formulations that have been studied with antidiabetic plant extracts or phytocompounds can be divided into vesicular and nonvesicular lipid carriers.

**Table 1 pharmaceutics-14-02135-t001:** Summary of the in vivo effects of current plant-based antidiabetic drugs or phytochemicals loaded in lipid-based nanoparticle carriers compared to unloaded drugs.

Type of Nanoparticulate Carrier	Extract/ Phytocompound Loaded	Size of the Nanocarriers	Model of the Diabetic Animal	Effects/Overcome	References
Liposomes	Betanin	40.06 ± 6.21 nm	Streptozotocin-induced rats	B cells protection, serum insulin levels increased, and glucose-lowering effects increased; 418.70 ± 31.38 mg/dL for diabetic rats, 390.16 ± 24.31 mg/dL for diabetic rats treated with free betanin, and 185.11 ± 27.27 mg/dL for rat treated with betanin liposomes	[193]
	Liquiritin	91.84 ± 1.85 nm	Streptozotocin-induced diabetic rats	8.8-fold bioavailability increased and high hypoglycemic and antioxidant effect than free liquiritin	[194]
	*Momordica charantia*, *Trigonella foenum-graecum*, and *Withania somnifera* freeze-dried alcoholic extract (50% *v*/*v*)	1176 ± 5.6 nm	Streptozotocin-induced diabetic rats	More than 2-fold increased efficacy: on days 7, 14, and 21 of the experiment, 500 mg/kg of encapsulated extracts lowered blood glucose 17.11%, 38.39%, and 52.11%, respectively, comparable to metformin (500 mg/kg) 21.56%, 43.92%, and 54.76%. The oral unencapsulated marketed polyformulation 1000 mg/kg decreased blood glucose levels by 12.66%, 31.29%, and 45.62%, respectively, compared to baseline	[195]
	Hydroalcoholic extract of *Pterocarpus marsupium*	ND	Alloxan-induced diabetic rats	High efficacy of the liposome formulation. A total of 50 mg/100 gm body weight per day orally for 7 days of the liposome formulation decreased blood glucose levels by 70% while the free extract decreased by 28%	[196]
Phytosomes	Berberine	165.2 ± 5.1 nm	db/db diabetic mice	3-fold bioavailability increase than free berberine	[197]
	Combined flavonoid-rich extract from fruits of *Citrullus colocynthis* (L.) *Momordica balsamina* and *Momordica dioica*	450 nm	Streptozotocin-nicotinamide-induced diabetic rats	Sustained release of the flavonoid High antidiabetic efficacy of 100 mg/kg per day of the phytosomes than 250 mg of the free methanolic extract	[198]
Niosomes	Lycopene	202 ± 41 nm	Alloxan-induced diabetic rats	Glucose-lowering effect increased with lycopene-niosomes treatment 269 ± 13.3 to 109.6 ± 11.2 mg/dL compared to free pure lycopene 265 ± 6.8 to 136.5 ± 7.8 mg/dL	[199]
	Alcoholic extract of *Gymnema sylvestre*	229.5 nm	Alloxan-induced diabetic rats	Glucose-lowering effect increased with the noisome Significant reduction in blood glucose levels by 400 mg/kg AUC 0–2 h: 75.58 ± 1.69 compared to the nonencapsulated extract AUC 0–2 h: 80.48 ± 2.31. *Gymnema sylvestre* niosomes and the extract reduced the blood glucose 24.01 ± 2.00% and 15.68 ± 2.06%, respectively, compared with its initial values	[200]
Solid lipid nanoparticles (SLNs)	Myricetin	76.1 nm	Streptozotocin-nicotinamide-induced diabetic mice	Decrease in blood glucose level by myricetin-SLNs 10 mg/kg was similar to that by metformin 200 mg/kg after 4 weeks of daily oral administration	[201]
Nanostructured lipid carriers (NLC)	Baicalin	92 ± 3.1 nm	Streptozotocin-induced diabetic rats	High antidiabetic efficacy compared with baicalin. Decreased fasting blood glucose level 20% more and glycosylated hemoglobine 16% more than free baicalin	[202]
Nanoemulsions (NEs)	Bitter gourd seed oil	93.9 ± 2.6 nm	Alloxan-induced diabetic rats	Glucose-lowering effects increased; Antioxidant effects increased (Values not given)	[172]
	Berberine	30.56 ± 0.35 nm	High-fat diet and streptozocin-induced diabetic mice	Bioavailability increased by 212.02%; glucose-lowering effects increased 3-fold; regulation of liver function	[203]
	Sage Essential Oil	143.2 nm	Alloxan-induced diabetic wistar rats	Glucose-lowering effects increased maximum decrease in blood glucose of 74.32% compared to free sage oil 50.45%. Regeneration of pancreatic tissue	[204]
Self-nano emulsifying drug delivery system (SNEDDS)	Polypeptide-k	31.89 nm	Streptozotocin-induced diabetic rats	Hypoglycemic effects rose >60%. Increased oxidation inhibiting effect; Hyperlipidemia preventing effects increased; Renewal of pancreatic tissue	[205]
	Curcumin	170 nm	Streptozotocin-induced diabetic rats	Significant decrease in tumor necrosis factor alpha (TNF-alpha) in diabetic neuropathy. ~10% more than free curcumin	[206]
	Resveratrol	336 ± 11.6 nm	Streptozocin diabetic-induced albino rats	Significant hypoglycemic and hypolipidemic effects at dose 10 mg/kg comparable to two time this amount with free resveratrol	[207]

ND: non-defined.

#### 4.1.1. Vesicular lipid nanocarriers used for antidiabetic plant extracts or phytocompounds

Vesicular lipid nanocarriers are systems in which an active drug is encapsulated in a vesicular structure composed of an aqueous phase surrounded by an oily shell. Hydrophilic agents are enclosed in the aqueous part, while lipophilic drugs are present in the oily moiety [208]. Vesicular delivery systems carrying antidiabetic herbal medicines that have been studied include liposomes, phytosomes, and niosomes (Figure 2).

##### Liposomes

Liposomes are stable particles generally composed of an aqueous core surrounded by a lipid bilayer envelope constituted mainly with amphiphilic phospholipids (with a hydrophilic head and a hydrophobic tail) [209]. Their lipid bilayer and their aqueous core may incorporate hydrophobic or hydrophilic compounds, respectively [210]. Their structure imitates that of cell membranes and promotes the entry of the drug into cells [211,212]. Studies have shown their benefits as carriers for enhancing the stability, bioavailability, and activity of plant extracts and isolated phytocompounds. As an example, Amjadi et al. prepared betanin liposomes with an encapsulation efficiency (EE) of 80.35% and a loading capacity (LC) of 26.78%. Betanin-loaded liposomes (20 mg/kg per day of betanin) were administered to diabetic rats; the results showed that betanin-loaded liposomes were significantly more active and positively regulated hyperglycemia, hyperlipidemia, and oxidative stress than free betanin. Liposomes improved the stability and antioxidant activity of betanin in in vitro studies. Tissue analysis showed that betanin loading was safer than free betanin loading [193]. Similar benefits have been shown by Wang et al. with liquiritin-loaded liposomes (EE 92.5%, 200 mg/kg per day) on rats with an increase in relative oral bioavailability of 8.8-fold, yielding great hypoglycemic and antioxidant effects in comparison to free liquiritin [194].

In a study by Gauttam and Kalia, liposome encapsulation of freeze-dried hydroalcoholic extracts of antidiabetic plants (*Momordica charantia*, *Trigonella foenum-graecum*, and *Withania somnifera* at a ratio of 2:2:1) was performed with an EE of 66.9%. Then, the antidiabetic potential was compared to that of the free-marketed polyherbal formulation in diabetic rats, which received the formulations once daily for 21 days. The study showed that the antidiabetic potential was enhanced by encapsulation in liposomes: 500 mg/kg exhibited a more significant potential (*p* < 0.05) compared to 1000 mg/kg of the free polyherbal formulation [195]. Liposomes encapsulating the hydroalcoholic extract of *Pterocarpus marsupium* (EE 82.7) were prepared by Singh et al. in their investigation. Then, they performed a blood glucose-lowering study on diabetic rats. The oral dose was 50 mg/100 g body weight per day for 7 days. At the end of the study, it was reported that the crude extract of *Pterocarpus marsupium* lowered blood glucose levels from 390.1 mg/dl (control) to 280.8 mg/dL, but the liposomes of *Pterocarpus marsupium* lowered blood glucose levels to 113.1 mg/dL. Thus, it was concluded that the liposomal formulation of *Pterocarpus marsupium* was more effective than the normal extract [196].

##### Phytosomes

Phytosomes are recently developed nanoparticulate systems that are produced from the reaction of polyphenolic constituents or standardized plant extracts (curcumin, berberine, methanolic extract of pomegranate peels, etc.) and the polar head regions of phospholipids (phosphatidylcholine, etc.) [213,214]. Phytosomes have been shown to increase the absorption of phytocompounds or plant extracts, resulting in improved biological activity. Berberine was administered to rats as berberine-phytosomes (EE 85%, single dose 50 mg/kg). The formulation increased the oral bioavailability by 3-fold compared to conventional oral administration of berberine and more effectively improved glucose metabolism [197]. In a pharmacokinetic study conducted by Riva et al., quercetin-phytosomes, 500 mg orally administered to human volunteers, were found to increase quercetin plasmatic levels up to 20 times more than free quercetin (EE not given). Additionally, no notable side effects were shown after examination of the participants [215].

Rathee and Kamboj prepared phytosomes of flavonoids from a methanolic extract of *Citrullus colocynthis* (L.), *Momordica balsamina*, and *Momordica dioica* fruits with EE 92.1± 5.1%. They compared their antidiabetic activity to that of free methanolic extract on rats after daily administration for 15 days. The treatment with the 100 mg/kg per day polyherbal phytosomal formulation resulted in a higher decrease in the levels of fasting blood glucose (from 276 to 148 mg/dL) than the free methanolic combined extract of all plants (from 271 to 176 mg/dL) at the dose of 250 mg/kg per day. An in vitro study showed a sustained release of 92% of the formulation in 12 h [198].

##### Niosomes

Niosomes are surfactant vesicles with a bilayer structure formed by the self-assembly of nonionic surfactants and cholesterol in an aqueous phase [216]. In 2017, PK et al. prepared lycopene niosomes (EE 62.8%) and reported that 200 mg/kg per day, orally given to diabetic rats for 14 days, showed more effectiveness than free lycopene. In comparison to the control (260.3 mg/dL), they decreased the blood glucose levels to 91.6 mg/dL and 100.5 mg/dL, respectively. Furthermore, lycopene niosomes showed sustained release, increased stability of lycopene for a prolonged time and reduced total cholesterol values to 108.5 mg/dL while free lycopene reduced the cholesterol values to 161.4 mg/dL relative to the control of 242.4 mg/dL [199]. In another study, Kamble et al. prepared *Gymnema sylvestre* extract-loaded niosomes with an EE of 85.3%. An in vitro study showed the progressive release of the phytocompounds (77.4% within 24 h). In an oral glucose tolerance test, the formulation (400 mg/kg) showed a significant reduction in blood glucose levels in diabetic rats, with an AUC 0–2 h value of 75.58 compared to the nonencapsulated extract, with an AUC 0–2 h value of 80.5. Niosomes showed greater antihyperglycemic activity than free extract [200].

#### 4.1.2. Nonvesicular Lipid Nanocarriers Used for Plant Extracts or Phytocompounds

##### Solid Lipid Nanoparticles (SLNs)

Solid lipid nanoparticles are aqueous colloidal systems made of solid lipids at room temperature and stabilized by nonionic emulsifiers [217]. SLNs have been used as transporters for poorly soluble antidiabetic phytocompounds. Xue et al. prepared berberin-solid lipid nanoparticles with EE 58% and 4.2% LC. The formulation orally administered at a dose of 100 mg/kg in mice significantly increased the bioavailability of berberine compared to the same dose of free berberine [218]. In another study, SLNs (EE 56.2% and 5.62% LC) containing myricetin were administered daily to mice at 10 mg/kg orally. This resulted in a significant decrease in blood glucose similar to mice treated with 200 mg/kg metformin [201].

##### Nanostructured Lipid Carrier (NLC)-Antidiabetic Plant Extracts or Phytocompounds

Nanostructured lipid carriers are a second generation of solid lipid nanoparticles in which small amounts of liquid lipids (oils) at room temperature are incorporated to produce structural rearrangements of the matrix [219,220]. They are mostly prepared by high-pressure homogenization or by the microemulsion method [190]. Recently, baicalin-loaded nanostructured lipid carriers were prepared by Shi et al. (with EE 85.29%) and were orally administered to rats (200 mg/kg). The formulation exhibited a sustained release of baicalin and significantly lowered blood glucose by approximately 27%, glycosylated hemoglobin, total cholesterol, and total triglyceride more than free baicalin administered alone [202]. In an in vivo biodistribution study of nanostructured lipid carriers loaded with quercetin (EE 89.3%), Liu et al. observed a sustained release of quercetin in mice and 1.57-, 1.51-, and 1.68-fold increases in relative availability in the lung, liver, and kidney, respectively, compared to an intact quercetin solution [221].

##### Nanoemulsions and Self-Nanoemulsifying Drug Delivery Systems (SNEDDSs)

Nanoemulsions (NEs) consist of nanosized oil droplets in the aqueous phase that form a homogeneous dispersion by means of surfactants [222]. With droplet sizes ranging from tens to several hundreds of nanometers, they are kinetically stable [222,223]. Nanoemulsions are actually being manufactured to improve the bioavailability of low-solubility active herbal medicines or phytocompounds. Antidiabetic herbal medicines entrapped in NEs have demonstrated improved bioavailability and a higher glucose-lowering effect and stability [203,204]. As an example, α-eleostearic acid (constituting 50% bitter gourd oil) entrapped in NEs (0.5 and 1%) was administered to diabetic rats daily for 28 days. This resulted in a significant blood glucose-lowering effect by NEs compared with free α-eleostearic acid, and the increased stability of α-eleostearic in nanoemulsions at the end of the 12-week storage period was observed [172]. Recently, Javadi et al. showed that plant extract (fenugreek extract, nettle extract, and cumin essential oil)-loaded NEs had suitable stability, and a study on L6 cells demonstrated that plant extract-NEs could diminish blood sugar [224]. In another study, the *Abelmoschus esculentus* ethanolic extract nanoemulsion was prepared and orally administered to diabetic mice at a dose of 400 mg/kg body weight for 14 days. It was found that extract-loaded NEs decreased blood glucose by 52.05%, while the free extract lowered blood glucose by only 39.32% [225].

Self-nanoemulsifying drug delivery systems (SNEDDSs) are structurally very close to nanoemulsions; they consist of approximately 30–60% surfactants and cosurfactants or cosolvents, less than 20% by weight of oils, and the remainder being the drug [212]. There is no water used in their preparation [226]. Naturally, in aqueous media, they form fine oil-in-water nanoemulsions with nanodroplets <200 nm in size [212,226]. Due to their composition, SNEDDSs are being used as nanoparticulate carriers for improving the bioavailability and stability of lipophilic drugs. Garg et al. loaded polypeptide-k in SNEDDSs and studied the dissolution of polypeptide-k. They found that all polypeptide-k was released within 15 min in SNEDDS, while only 18.42% was released in 1 h with the free drug. The dissolution rate was improved 5.4-fold by SNEDDSs. In the same study, administration of 800 mg/kg daily polypeptide-k in SNEDDSs for 28 days decreased glycemia in diabetic rats to less than 100 mg/dL, while the glycemia of the rats receiving the same dose of free drug remained similar to that of the control group, slightly more than 250 mg/dL [205]. A study by Rayanta et al. reported that curcumin SNEDDSs at a dose of 250 mg/kg were administered orally to rats. Investigation of its bioavailability revealed that the maximum concentration and the area under the curve in the blood improved by 1632.1% and 7411.1%, respectively. In the same study, in diabetic neuropathy in rats, high doses (100 and 300 mg/kg/day orally for 2 weeks) of curcumin SNEDDSs and free curcumin produced an amelioration in tail flick latency, but the SNEDDSs formulation improved more than the free curcumin [206]. An in vivo study by Khursheed et al. showed that the bioavailabilities of curcumin and quercetin increased approximately 50.2- and 5.6-fold, respectively, when used in SNEDDS [227]. In another study, Balata et al. prepared resveratrol SNEDDSs and showed that their dissolution efficiency was higher (94%) than that of pure resveratrol (42%). Then, an in vivo antidiabetic study in diabetic rats showed that daily administration of 10 mg/kg resveratrol SNEDDSs for 4 weeks exhibited significant hypoglycemic and hypolipidemic effects similar to those of 20 mg/kg free resveratrol [207]. In a study of stability, SNEDDSs of ethanolic leaf extract of *Ipomoea reptans* as antidiabetic treatment were prepared by Jumaryatno et al. Stability tests, including centrifugation, heating-cooling, freeze–thaw and endurance tests, were conducted and did not show phase separation or modification of particle size [228]. Hayati et al. studied the effectiveness of SNEDDS of an ethanolic extract of *Centella asiatica* (L.) leaf (a plant used to treat diabetes in Indonesia) to reduce fasting blood glucose levels in zebrafish. After inducing diabetes by alloxan and glucose, the treatment consisted of soaking the zebrafish in solutions of 25 mg/2 L metformin or 100 mg/2 L or 200 mg/2 L SNEDDS. The antidiabetic activity test showed reductions in blood glucose of 65.5% by metformin, 69.9% by 100 mg/2 L SNEDDS and 72.2% by 200 mg/2 L SNEDDS [229]. A comparison with free extract was not performed. Similar results were obtained by the same author using *Ipomoea reptans* ethanolic extract-SNEDDSs [230].

### 4.2. Inorganic-Based (Metal or Metal Oxide) Nanoparticle Formulations for Antidiabetic Plant Extracts or Phytocompounds

Metal or metal oxide nanoparticle formulations that have been prepared with antidiabetic plant materials are also called green-synthesized metal or metal oxide nanoparticles. Green-synthesized preparations are made by the reduction in metal ions utilizing bioactive agents of plant materials [231,232]. They are generally prepared by mixing the metal precursor salt preparation with the plant extract solution. During the process, metallic ions are reduced and form metal or metal oxide nanoparticles surrounded by a mixture of phytochemicals. Then, they are collected and dried [233,234,235]. Their potential application against diabetes has recently driven great interest. Their inorganic part (i.e., the metal or metal oxide) functions as a carrier of phytoconstituents and not only increases their antidiabetic effect but also acts synergistically [236,237]. The green synthesis strategy is currently emerging to deliver not only the phytochemicals but also the metallic element on which they are attached [9]. Thus, inorganic-based nanoformulations of herbal products offer certain benefits to the phytochemicals attached, including the ability to enter the systemic circulation and reach deeper organs. Furthermore, the biological activity of green-synthesized nanoparticles can depend on the applied components [10].

Some inorganic elements, such as silver, zinc, selenium, gold, iron, platinum, manganese, magnesium, copper, chromium, vanadium palladium, nickel, titanium, molybdenum, cerium, and tungsten, have exhibited in vitro and/or in vivo antidiabetic activities by various mechanisms, including being cofactors of enzymes and increasing glucose utilization, insulin sensitivity and antioxidant enzyme levels [238,239,240,241,242,243]. Nanoparticles have been biologically fabricated with several of them by plant mediation, namely, green synthesis, and have shown improvements in antidiabetic effectiveness compared to only metals or plants [244] and chemically or physically synthesized nanoparticles [245]. Nevertheless, very few studies have compared their in vitro and in vivo effectiveness with those of their bulk counterparts.

#### 4.2.1. Green-Synthesized Silver Nanoparticles

Green-synthesized silver nanoparticles (AgNPs) are generally obtained by mixing plant extracts and silver salts. Due to its high solubility, silver nitrate (AgNO_3_) is the most commonly used salt compared to sparingly soluble silver salts (AgCl, AgBr, AgI and Ag_2_S), which leads to a slower release of silver ions during synthesis. [235,246,247]. They have been reported to possess in vitro and in vivo antidiabetic activity (Table 2). Balan et al. demonstrated suitable antidiabetic ability of AgNPs fabricated with aqueous leaf extract of *Lonicera japonica.* In vitro assessment of the synthesized nanoparticles showed remarkable inhibition of carbohydrate digestive enzymes and were identified to be reversible noncompetitive inhibitors for key enzymes of diabetes (α-amylase and α-glucosidase) [248]. Similarly, *Allium cepa* extract-AgNPs were synthesized and tested in vitro; the results showed significant inhibition of carbohydrate digestive enzymes (α-glucosidase and α-amylase) comparable to acarbose and free radical-scavenging ability (Jini and Sharmila, 2020). In vivo studies of *Musa paradisiaca* stem extract-mediated synthesized silver nanoparticles on diabetic rats have also shown suitable antidiabetic effects by decreasing the levels of blood glucose and glycosylated hemoglobin [249]. Similar effects have been reported using *Solanum nigrum* leaf extract-AgNPs [250] and *Zingiber officinales* extract-AgNPs [251].

**Table 2 pharmaceutics-14-02135-t002:** Overview of the antidiabetic activities of inorganic-based nanoformulations of antidiabetic plant products.

Inorganic Carrier	Plant Extract Used	Study Considered	Size	Findings/Outcomes	References
Silver nanoparticles (AgNPs)	*Lonicera japonica* aqueous leaves extract	In vitro α-amylase, and α-glucosidase inhibitory activity	20–60 nm	Significative potential antidiabetic activity against the two enzymes with inhibitory concentration (IC50) of 54.56 µg/mL for α-amylase and 37.86 µg/mL for α-glucosidase	[248]
	Aqueous extract of the leaves of *Calophyllum* *tomentosum*	α-amylase, α-glucosidase, and dipeptidyl peptidase IV (DPPIV) inhibition assay	24 nm	Strong inhibition of gastric (α-glucosidase enzyme) and incretin inhibitor (DPPIV enzyme) compared to the pancreatic one (α-amylase enzyme)	[252]
	*Musa paradisiaca* aqueous stem extract	In vivo antidiabetic potential in male albino rats of Sprague–Dawley strain	30–60 nm	Decrease in the levels of blood glucose by 26% by a dose of 50 µg/kg, similar to that of 600 µg/kg glibenclamide (28%), a similar increase in insulin level rat receiving nanoparticles and those receiving glibenclamide and glycogen levels	[249]
	*Eysenhardtia polystachya* aqueous bark extract	in vivo antidiabetic activity assay in Zebra fish, insulin secretion assay	10–12 nm	Suitable antidiabetic activity: 5 µg/mL decreased the glycemia by 38.1% and a decrease by 44.5% was observed by 10 µg/mL, in comparison to 51% by 5 µg/mL glibenclamide Significant increase in insulin secretion, elevation of insulin receptor expression isoforms	[253]
	*Solanum nigrum* methanolic leaves extract	In vivo Oral glucose tolerance test in wistar albino rats	4–25 nm	Significant decrease in the blood glucose level by 10 mg/kg 2 h after (comparable with glibenclamide) and improvement in the body weight after 21 days	[250]
	*Zingiber officinales* ethanolic extract	In vivo antidiabetic activity in streptozocin-induced diabetic rats	123.8 nm	Decreasing of blood glucose level to normal from 249 ± 0.67 mg/dL to 86 ± 0.91 mg/dL on 7th day after the administration by 200 mg/kg of body weight similar to metformin 82 ± 1.9 mg/dL	[251]
Zinc oxide nanoparticles (ZnONPs)	*Azadirachta* *indica*, *Hibiscus rosa-sinensis*, *Murraya koenigii*, *Moringa oleifera*, and *Tamarindus indica* aqueous leaves extract	In vitro α-amylase and α-glucosidase inhibitor activity and antioxidant activity	27–54 nm	Exhibition of appreciable α-amylase and α-glucosidase inhibitory activity compared with chemically synthesized ZnO nanoparticles and enhanced antioxidant activity contrarily to chemically synthesized ZnO nanoparticles	[254]
	*Hibiscus subdariffa* leaf aqueous extract	In vivo antidiabetic activity on streptozocin-induced diabetic mice	12–46 nm	Suitable antidiabetic effect of ZnONPS (at a dose of 8 mg/kg per day for a period of 28 days): reduction in glycemia in mice treated up to 59.58% than that of untreated	[255]
	*Vaccinium arctostaphylus* L fruit ethanolic extract 96%	In vivo antidiabetic effect on alloxan-induced diabetic rats	15 nm	High efficacy of biosynthesized ZnONPs (8 mg/dL). Most effective reduction in fasting blood glucose to 50.4 ± 3.55 than the extract (150 mg/dL) 79.4 ± 2.85 more than chemically synthesized (8 mg/dL) 83.2 ± 9.54 more than insulin (10 UI/kg) 113.8 ± 12.96 in treating the alloxan-diabetic rats daily for 16 days compared to diabetic non-treated rats 199.8 ± 10.94 mg/dL	[244]
	*Silybum marianum* L aqueous seeds extract	In vivo antidiabetic activity on alloxan-induced diabetic wistar rats	19.9 nm	Decrease in fasting blood sugar from 207 mg/dL to 96 mg/dl by 10 (mg/dL) of green-synthesized ZnO after 16 days of treatment more than the extract (150 mg/dL) and then the insulin 10 UI/kg and insulin level increased significantly to 1,7 µg/dL compared to rat receiving chemical ZnONPS and the extract (about 0,9 µg/dL)	[237]
	*Costus igneus* aqueous leaves extract	In vitro α-amylase and α-glucosidase inhibition assays	26.55 nm	Suitable antidiabetic and antioxidant activity α-amylase and α-glucosidase inhibition activity of synthesized zinc oxide nanoparticles was 74% and 82%, respectively, and the 2,2-diphenyl-1-picrylhydrazyl hydrate reducing power at 100 μg/mL was 75%	[256]
	*Eryngium billardieri* aqueous Leaf Extract	In vivo alloxan-induced diabetic rats antidiabetic activities	34 nm	The green ZnONPs 7 mg/kg daily for 16 days showed an excellent efficiency in overcoming diabetic rise of high-density lipoprotein and greatly reduced fasting blood sugar levels to 103.2 mg/dL compared to the extract alone 116.9 mg/dL and the chemically synthesized zinc oxide nanoparticles 114.1 mg/dL, cholesterol reduction, and increased insulin levels 1.29 µg/dL than the extract 0.48 µg/dL and the chemically synthesized zinc oxide nanoparticles 0.48 µg/dL	[245]
Selenium nanoparticles (SeNPs)	*Hibiscus sabdariffa* aqueous leaf extract	Evaluation of factors correlated to oxidative damage in in vivo streptozocin-induced diabetic Wistar rats	ND	1 mg of synthesized SeNPs/10 kg body weight, given once per day for a period of 28 days elevated antioxidant enzyme activities and the glutathione value (to more than 0.35 mmol/mg protein) in testicular tissues than insulin treatment (less than 0.25 mmol/mg protein)	[257]
	Mulberry leaf and *Pueraria lobata* ethanolic extracts	In vivo hypoglycemic effect in diabetic rats	120 nm	Significant hypoglycemic effects with decrease in blood glucose up to 32.88% compared to plant extracts and nanoparticles alone at a dose of 125 mg/kg. Synergistic effect between plant extracts and SeNPs	[258]
	*Catathelasma ventricosum* polysaccharides solution	Antidiabetic activity in streptozocin-induced diabetic male outbred stock from Institute of Cancer Research (ICR) mice	50 nm	Higher antidiabetic activity than chemical selenium NPs, selenium salt, and the extract alone 2 mg/kg daily for 30 consecutive days. Decrease in glycemia from 24 to 8.3 mmol/L, 24.4 to 17.7 mmol/L, and 24.4 to 12.6 mmol/L, respectively Synergistic effect between plant extracts and SeNPs. Improvement in body weight, blood sugar, antioxidant enzymes activities, and lipid levels	[259]
Gold nanoparticles (AuNPs)	*Leucosidea sericea* procyanidins fractions	In vitro α-amylase and α-glucosidase inhibitory activity	6–24 nm	Stronger α-amylase inhibitory activity (IC50 value at 1.88 µg/mL) and α-glucosidase activity (IC50 value at 4.5 µg/mL) than the fractions alone (IC50 value at 3.5 ± 0.7 µg/mL and 8.1 ± 0.6 µg/mL, respectively), more efficient than acarbose (610 ± 2.6 µg/mL and 10.2 ± 0.6 µg/mL, respectively)	[260]
	*Eclipta alba* methaolic extract	In vitro evaluation of prevention of β-cell damage (RIN-5F pancreatic β-cells)	26 nm	Anti-apoptotic potential of β-cell decreased cell damage. Decreased level of Bcl-2 protein and increased level of Bax indicates the cell survival induced by NF-κB pathway	[261]
	*Fritillaria cirrhosa* aqueous extract	In vivo antidiabetic effect in streptozotocin diabetic rats	40–45 nm	20 mg/kg per day for 28 days decreased blood glucose to just under 150 mg/dL in comparison to controlling glycemia by about 300 mg/dL and increased insulin level. A decrease in glycosylated hemoglobin, modulation of antioxidants level, decrease in lipid peroxidation, and regeneration of islets cells were seen	[262]
	*Cassia auriculata* propanoic acid 2-(3-acetoxy-4,4,14-trimethylandrost-8-en-17-yl) (PAT)	In vivo antidiabetic activity in alloxan-induced diabetic rats	12–41 nm	Green-synthesized gold nanoparticles (0.5 mg/kg daily for 28 days) significantly decreased the plasma glucose level to lower than 150 mg/dL in comparison to diabetic control, which was more than 300 mg/dL. Cholesterol levels, as well as triglyceride levels, decreased significantly, and an elevation of insulinemia was shown with synthesized nanoparticles	[263]
	*Sargassum swartzii* extract	In vivo antidiabetic effect using male wistar Albino rats	37 nm	Green-synthesized gold nanoparticles (0.5 mg/kg per day orally for 28 days) significantly decreased fasting blood glucose levels to 98 ± 7.2 mg/dl compared to diabetic control 218 ± 9.6 mg/dL. Glycosylated hemoglobin levels decreased to 0.34±0.041 mg/g Hb compared to diabetic control 0.84 ± 0.05. Insulin levels increased in the diabetic rats receiving green-synthesized gold nanoparticles	[264]
	*Cassia fistula* stem bark aqueous extract	In vivo antidiabetic streptozotocin-induced diabetic rats	55.2–98.4 nm	Green-synthesized gold nanoparticles (60 mg/kg bw oral gavage) decreased serum blood glucose concentrations to 168.47 ± 16.18 mg/dL by synthesized nanoparticles in comparison to rats treated with the extract (60 mg/kg bw) 211.05 ± 5.40 mg/dL and HbA1c levels to 10.40% ± 0.23% and 11.45% ± 0.28%, respectively. Improvement in body conditions and the activity of transaminase enzymes, Upregulation of lipid profile, reversion of kidney failure	[265]
	*Moringa oliefera* leaves aqueous extract	In vitro α-amylase inhibition activity	15.2 nm	Inhibition of the enzyme in a concentration-dependent mode IC50 value of 130 µg/mL	[266]
Copper nanoparticles (CuNPs)	*Dioscorea bulbifera* extract	In vitro α-glucosidase inhibition activity	12–16 nm	99.1 ± 0.2% and 90.7 ± 0.3% suppression of α-glucosidase and murine intestinal glucosidase activity, respectively	[239]
	*Millettia pinnata* aqueous extract of the flower	In vitro α antidiabetic activity	23 ± 1.10 nm	Inhibition of α-amylase in a dose-dependent manner Constrained glycosylation at 77% compared with the extract solution (54%) High antioxidant property offered by extract functional groups	[267]
	*Gnidia glauca* leaves aqueous extracts	In vitro α glucosidase inhibition assay	70–93 nm	Inhibition of 88.60 ± 0.78% at 100 μg/mL	[268]
	*Bacopa monnieri leaf extract*	In vivo antidiabetic effect in streptozocin-induced diabetic mice	34.4 nm	Single dose of synthesized CuNPs at 14 mg/kg by oral route and 7 mg/kg orally accompanied by subcutaneous 0.2 U of insulin per 50 g decreased blood glucose levels by almost 33.7 and 32.2%, respectively	[269]
Iron nanoparticles (FeNPs)	Polyherbal Formulation Containing *Tinospora cordiofolia*, *Curcuma longa Trigonella foenum gracum, Emblica officinale*, and *Salacia* *oblonga* methanolic extract	In vitro α-amylase inhibitory assay	40–60 nm	Equivalent α amylase enzyme inhibitory activity (70.48%) to the standard ascorbic acid (73.87%) at the maximum concentration of 250 µg/mL	[270]
	*Sesamum indicum* aqueous seeds extract	In vitro α-amylase inhibitory assay	99 nm	α-amylase inhibition activity was shown in a dose-dependent manner (10–100 µg/mL). The highest inhibitory activity, was 64.39 ± 0.52% with IC50 value = 21.26 μg/mL	[271]
Platinum nanoparticles (PtNPs)	*Polygonum salicifolium* leaves aqueous extract	In vitro α-amylase and α-glucosidase inhibition assay	1–3 nm	Remarkable suppression of α-glucosidase activity with IC50 = 53 μg/mL in comparison to that of acarbose and curcumin and slight α-amylase inhibitory effect	[272]
	*Whitania somnifera* leaves	In vivo antidiabetic activity in streptozotocin-induced diabetic rats	12 nm	1 mg/kg PtNPs decreased plasma glucose levels to the normal level (117.34 ± 4.18 mg/dL) than intraperitoneal injection of 10 mL/kg plant extract (153.34 ± 8.16 mg/dL) compared to the control group (453.34 ± 8.16 mg/dL) after 28 days treatment	[273]
Platinum-titanium oxide nanoparticles (Pt-TiO_2_NPs)	*Costus speciosus* 99% ethanolic leaf extract	α amylase inhibition test	ND	Suitable α-amylase inhibition effect. *Costus speciosus*-Pt-TiO_2_NPs at 1% and 3% showed α-amylase inhibition of 71.1 ± 1.0% and 77.4 ± 1.7%, respectively, at a dose of 100 µg/mL	[274]
Titanium dioxide nanoparticles (TiO_2_NPs)	*Azadirachta indica* leaves aqueous extract	α-amylase inhibition assay	ND	Up to 97.2% inhibition was produced by TiO_2_NPs at 50 µg/mL, which was similar to the effect exhibited by acarbose	[275]
	*Stevia rebaudiana (Bertoni)* Bertoni leaves alcohol-aqueous (80:20) extract	Hypoglycemic effect in alloxan-induced diabetic rats	100 nm	Induction of strong and sustained reduction in the glucose level up to 30 days after green-synthesized TiO_2_NPs administration at a dose of 1 g/kg intraperitoneally (solution 20 and 30 μM). The glycemia was lowered by 20% and 56%, and glycated hemoglobin by 25% and 41% compared to the reduction by the chemically synthesized TiO_2_NPs	[276]
Palladium oxide nanoparticles (PdONPs)	*Zanthoxylum armatum* fruit aqueous extract	α-glucosidase inhibition assay	10.5 nm	Excellent enzyme inhibition with IC50 = 0.0218 ± 0.01 μg/mL	[277]
Nickel oxide nanoparticles (NiONPs)	*Areca catechu* leaf extract	α-amylase inhibition assay	5.46 nm	NiONPs showed an inhibition of the enzyme with IC 50 value = 268.13 µg/mL	[278]
Cerium oxide (CeO), zinc oxide, and silver nanoparticles	*Momordica charantia* fruits extract	In vivo antidiabetic effect in streptozotocin-induced diabetic rats	24 nm	200 mg/kg per day for 15 days of synthesized nanoparticles: cerium oxide, zinc oxide, and silver decreased blood glucose to 179.78 ± 5.12 mg/dL, 132.07 ± 6.21 mg/dL and 103.07 ± 6.21 mg/dL, respectively, and extract to 214.98 ± 3.43 mg/dL in comparison to the control 268.89 ± 3.76 mg/dL	[243]
Cerium oxide nanoparticles (CeONPs)	fruit extract of *Morus nigra*	In vitro glucose uptake of L6 cell lines	8.5 nm	100 μg/mL CeONPs enhanced glucose uptake to more than 65% higher than untreated cells	[279]

ND: non-defined.

#### 4.2.2. Green-Synthesized Zinc Oxide Nanoparticles

The synthesis of zinc oxide nanoparticles by the plant route is usually performed by mixing zinc nitrate hexahydrate (Zn(NO_3_)_2_.6H_2_O), zinc acetate (Zn(OOCCH_3_)_2_), or zinc sulfate (ZnSO_4_) with plant extracts [280,281]. The pH and the temperature of the medium are sometimes increased to approximately 8 to 10 and 70 °C, respectively, to improve the synthesis [234,282,283]. Zinc is a crucial trace element for animal cells and plays a significant role in the treatment of diabetes. Zinc significantly contributes to the improvement in fasting and postprandial blood sugar levels and the reduction in glycated hemoglobin [284]. It increases the synthesis, storage, and release of insulin and superoxide dismutase activity, reduces proinflammatory cytokine activity, inhibits alpha glucosidase and alpha amylase, and increases hepatocyte activity and lipid metabolism in the liver [242,285]. Zinc oxide nanoparticles are currently used in many consumer products and are considered GRAS [286,287] Kazempour et al. reported that green-synthesized zinc oxide nanoparticles using *Eryngium billardieri* leaf extract and zinc nitrate solution showed, in diabetic rats, a greater decrease in blood glucose levels and a significant insulin level increase due to the presence of phytocompounds. Furthermore, the green-synthesized ZnO showed high cholesterol reduction compared to rats treated with the *Eryngium billardieri* leaf extract and insulin [245]. Similar results were obtained in another study using *Silybum marianum* L. seed extract [237]. Antidiabetic plants, including *Azadirachta indica*, *Hibiscus rosa-sinensis*, *Murraya koenigii*, *Moringa oleifera*, *Tamarindus indica* [254], and *Hibiscus subdariffa* [255], have mediated zinc oxide nanoparticle biosynthesis, which showed very significant antidiabetic effects.

#### 4.2.3. Green-Synthesized Selenium Nanoparticles

Green-synthesized selenium nanoparticles are generally obtained after combining sodium selenite salt (Na_2_SeO_3_) [288] or selenious acid (H_2_SeO_3_) [289,290,291] with plant extracts. They have shown antidiabetic activities. Green-synthesized SeNPs prepared by Fan et al. by using *Hibiscus sabdariffa* leaf extract and selenious acid showed in vivo antidiabetic activities and protected against oxidative stress [257]. In another study, SeNPs were synthesized using ethanolic extracts of mulberry leaf and *Pueraria lobata*. In the simulated digestive fluid, they exhibited slow drug release and suitable stability. After oral administration in rats, significant hypoglycemic effects in both normal and diabetic rats were observed. Reduced oxidative stress improved pancreatic function, and glucose utilization by adipocytes was also reported [258]. Selenium nanoparticles synthesized using *Catathelasma ventricosum* polysaccharides have shown significantly higher antidiabetic activity than selenium nanoparticles synthesized chemically in streptozocin-induced diabetic male mice [259].

#### 4.2.4. Green-Synthesized Gold Nanoparticles

Green-synthesized gold nanoparticles are currently fabricated by reacting phytochemicals in plant extracts with gold ions provided by a precursor such as sodium tetrachloroaurate (III) dihydrate, choloroauric acid (HAuCl_4_), and auric chloride (AuCl_3_) [260,261,262,263].

Badeggi et al. synthesized gold nanoparticles using procyanidin fractions from the *Leucosidea sericea* total extract. In vivo studies resulted in higher α-glucosidase and α-amylase than the procyanidin fraction alone and acarbose, the most commonly used α-glucosidase and α-amylase inhibitor [260]. Gold nanoparticles synthesized using *Cassia auriculata* were administered in alloxan-induced diabetic rats at a dose of 0.5 mg/kg body weight by Venkatachalam and coworkers. These green-synthesized nanoparticles were demonstrated to decrease blood glucose levels together with cholesterol and triglycerides *(p* < 0.001) [263]. In another study, Opris et al. synthesized gold nanoparticles with *Sambucus nigra* L. extract and screened their antidiabetic activity in streptozocin-induced diabetic rats. For 2 weeks, a single oral dose of 15 mg/kg body weight of extract and 0.3 mg/kg body weight of synthesized gold nanoparticles were administered daily. This treatment resulted in decreases in blood glucose levels, inflammatory factors, and oxidative stress induced by hyperglycemia, which was higher than that induced by the extract. No toxicity was observed. The study showed that a low dose of green-synthesized gold nanoparticles could be used in diabetic therapy due to the increase in antioxidant defense and the reductions in metalloproteinase activity and inflammation in liver tissue [292]. In another study, *Fritillaria cirrhosa* aqueous extract mediated the synthesis of gold nanoparticles. The preparation exerted in vivo antidiabetic properties in diabetic mice and reduced serum markers (aspartate transaminase (AST), alanine transaminase (ALT), and alkaline phosphatase (ALP)) to the normal ranges when compared to those in diabetic rats [262]. A summary of the antidiabetic effects of green-synthesized gold nanoparticles is shown in Table 2.

#### 4.2.5. Green-Synthesized Iron Nanoparticles

Iron nanoparticles (Fe_2_O_3_) and (Fe_3_O_4_) are currently green-synthesized by the bioreduction of Fe(II) or Fe(III) salts such as ferrous sulfate (FeSO_4_), ferric chloride hexahydrate (FeCl_3_·6H_2_O), ferric nitrate nonahydrate (Fe(NO_3_)_3_.9H_2_O), and ferric acetylacetonate (Fe(C_5_H_8_O_2_)_3_) using plant extracts [271,293,294]. Green-synthesized iron nanoparticle antidiabetic activities are currently being assessed by researchers. Iron nanoparticle synthesis has been mediated by a polyherbal formulation containing *Tinospora cordiofolia*, *Curcuma longa*, *Trigonella foenum gracum*, *Emblica officinale*, and *Salacia oblonga* methanolic extracts. The antidiabetic activity was studied using an α-amylase assay and free radical-scavenging activity. This resulted in 70.5% α-amylase inhibition and a significant percentage of inhibition (radical-scavenging activity) of 74.5% at 250 μg/mL, which indicated that green-synthesized FeNPs possess better antioxidant and antidiabetic activity and can be used efficiently for the management of hyperglycemia [270]. Bano et al., in another investigation using *Sesamum indicum* aqueous seed extract as a bioreducer of iron, reported significant α-amylase inhibitory activity (more than 50% inhibition) of green-synthesized iron nanoparticles [271].

#### 4.2.6. Green-Synthesized Copper Nanoparticles

Copper is an essential element and plays important biochemical roles in growth, cardiovascular integrity, neuroendocrine function, reactive oxygen species detoxification, and iron absorption [295]. Green-synthesized copper nanoparticles can be made by reacting copper salts such as cupric chloride (CuCl_2_), cupric acetate Cu(CH_3_COO)2, cupric sulfate (CuSO_4_), and cupric nitrate (Cu(NO_3_)_2_) with phytoconstituents from plant extracts [268,269,296,297]. Studies have reported the efficient effects of green-synthesized copper nanoparticles in the treatment of diabetes through the inhibition of α-amylase and α-glucosidase activities and antioxidant activity [241]. Copper nanoparticles have been biosynthesized using extracts of plants such as *Gnidia glauca* and *Plumbago zeylanica* [268], *Dioscorea bulbifera* [239], and *Bacopa monnieri* [269] and have shown in vitro and/or in vivo antidiabetic activity at low doses.

#### 4.2.7. Green-Synthesized Cerium Nanoparticles

Green-synthesized cerium nanoparticles have been fabricated by mixing cerium(III) nitrate (Ce(NO_3_)_3_) salt or cerium(III) chloride (CeCl_3_) with plant extracts [298,299]. Shanker et al. synthesized cerium nanoparticles using *Momordica charantia* fruit extract and cerium nitrate solution. The green-synthesized cerium oxide nanoparticles showed higher antihyperglycemic activity than *Momordica charantia* extract alone [243]. Rajan et al. assessed the in vitro antidiabetic activity of green-synthesized cerium oxide nanoparticles by using L6 cell lines. The synthesized cerium oxide nanoparticles at a concentration of 100 μg/mL enhanced glucose uptake by more than 65% [279]. Additionally, Khan et al. reported that green-synthesized cerium nanoparticles can be considered effective antioxidant agents that can prevent oxidative damage to pancreatic islets due to various phytochemicals fixed on their surfaces [300].

#### 4.2.8. Green-Synthesized Platinum Nanoparticles

Green-synthesized platinum nanoparticles can be produced by reacting plant extracts with platinum ions from precursors such as sodium tetrachloroplatinate (II) (Na_2_PtCl_4_) or chloroplatinic acid hexahydrate (H_2_PtCl_6_.6H_2_O) [297,301]. Hosny et al. green fabricated platinum nanoparticles using *Polygonum salicifolium* leaf aqueous extract and platinum chloride solution (H_2_PtCl_6_⋅6H_2_O). In vitro studies of the produced nanoparticles exhibited suitable radical-scavenging and antidiabetic activity [272]. Li et al. fabricated platinum nanoparticles using *Whitania somnifera* leaves and chloroplatinic acid. Then, the evaluation of their effects in STZ-induced rats showed a higher blood glucose-reducing effect than the extract alone [273].

#### 4.2.9. Green-Synthesized Titanium Nanoparticles

Titanium nanoparticles have been produced by using titanium n-butoxide (Ti(OCH_2_CH_2_CH_2_CH_3_)_4_), metatitanic acid (TiO(OH)_2_), titanium dioxide powder (TiO_2_), or titanium tetraisopropoxide (Ti[OCH(CH_3_)_2_]_4_) as a source of titanium ions and mixed with plant extracts [275,276,297,301]. Samyuktha et al. prepared them using *Azadirachta indica* leaf aqueous extract and titanium dioxide powder. In vitro studies showed similar alpha amylase inhibitory activity with acarbose using the same dose [275]. In another study, Langlea et al. synthesized titanium nanoparticles by using titanium n-butoxide and *Stevia rebaudiana* leaf alcohol-aqueous (80:20) extract. Their investigations on diabetic rats showed powerful and prolonged antidiabetic activity. Thus, they concluded that TiO_2_ can be considered a suitable vehicle for the continuous release of active compounds in the treatment of diabetes [276].

#### 4.2.10. Other Green-Synthesized Inorganic Nanoparticles

Green palladium nanoparticles are produced using a palladium ion precursor such as palladium chloride (PdCl_2_) or palladium acetate (Pd(OAc)_2_) [297,301]. When the precursor is reacted with the plant extract, the phytocompounds act as bioreductors for the formation of these nanoparticles. Hazarika et al. showed the alpha glucosidase inhibitory activity of palladium nanoparticles synthesized with fruit extract of *Zanthoxylum armatum* [277]. Regarding nickel oxide nanoparticles, nickel salts such as nickel nitrate hexahydrate Ni(NO_3_)_2_·6H_2_O) or nickel chloride hexahydrate (NiCl_2_·6H_2_O) have been used as nickel ion precursors [297,301]. Shwetha et al. synthesized nickel oxide nanoparticles using *Areca catechu* leaf extract and nickel nitrate hexahydrate. The green-synthesized nanoparticles exhibited suitable alpha amylase inhibitory activity in vitro [278].

## 5. Discussion and Conclusions

Herbal products have been reported in the literature to be predominantly used in low-income countries for the treatment of various diseases, including diabetes. Various studies conducted on the compounds extracted from plants used in traditional medicine in the treatment of diabetes have shown their advantages as well as their limitations. Advantages such as the production of these compounds by several medicinal plants found in various regions of the world and multiple actions, including considerable antidiabetic activity, have been reported for various plant extracts. Despite these advantages, limitations have been reported, the main one being poor bioavailability of most active phytocompounds after oral administration due to either low solubility or permeability or their sometimes-rapid elimination. In addition, although medicinal plants are used in health care or commonly consumed as food, the compounds they contain have not been extensively studied for their long-term safety in chronic diseases. For chronic conditions, large amounts of phytocompounds must be ingested over time, and their risk is still poorly understood. To overcome these limitations and problems, strategies for improved formulations and reduced doses to be ingested with increased efficacy are underway. Indeed, these reported and ongoing strategies have shown that plant products (crude extracts or isolated compounds) incorporated into nanoformulations (either encapsulated within, through, or attached to the nanocarrier) are largely more effective on diabetes than when in unencapsulated form. Lipid nanocarriers, namely liposomes, solid lipid nanoparticles, and nanostructured lipid carriers, have shown significant encapsulation efficiency or loading capacity of plant products, which can make them preferred carriers for large amounts of drugs. The reported studies have shown that increased activity can be achieved with a low dose of drug carried by lipid nanocarriers. We noticed that these studies focused a lot on the evaluation of activity versus dose but did not address safety since the components of the lipid carriers are GRAS. However, considering a foreseen clinical application, this aspect must also be addressed. Regarding green-synthesized inorganic nanoparticles, they are composed of metallic elements of interest in the restauration of glycemic balance accompanied by phytochemicals. The studies did not quantify these phytochemicals but showed that the antidiabetic activity is considerably higher at low doses than that of non-green-synthesized inorganic nanoparticles. Nevertheless, these studies provided almost no information on their safety. Moreover, analyzing the reported works, almost all biosynthesized inorganic nanoparticles were produced using crude extracts, and very few studies evaluated the antidiabetic activity of green inorganic nanoparticles synthesized with isolated phytocompounds. In addition, many of these studies have not yet undergone sufficient in vivo experiments to better understand their effects and safety. ZnONPs are an exception due to their consideration as GRAS and their use as an additive in food products. The activity of zinc against diabetes is already known, and the results from in vivo studies point them as promising in foreseen clinical application, but this needs yet to be confirmed.

Finally, most antidiabetic studies conducted in vivo have been performed in chemically induced diabetic models (e.g., streptozocin or alloxan), which are models that primarily reflect type 1 but not type 2 diabetes. We suggest, e.g., the use of diabetic animal models induced naturally by diets to get much closer to the reality of type 2 diabetes (chronic hyperglycemia and insulin resistance) for a suitable analysis and interpretation of the results.

Altogether, based on the literature, nanoparticle-based delivery systems of plants and their compounds represent a promising alternative treatment in the management of diabetes, especially in low-income countries.

## Figures and Tables

**Figure 1 pharmaceutics-14-02135-f001:**
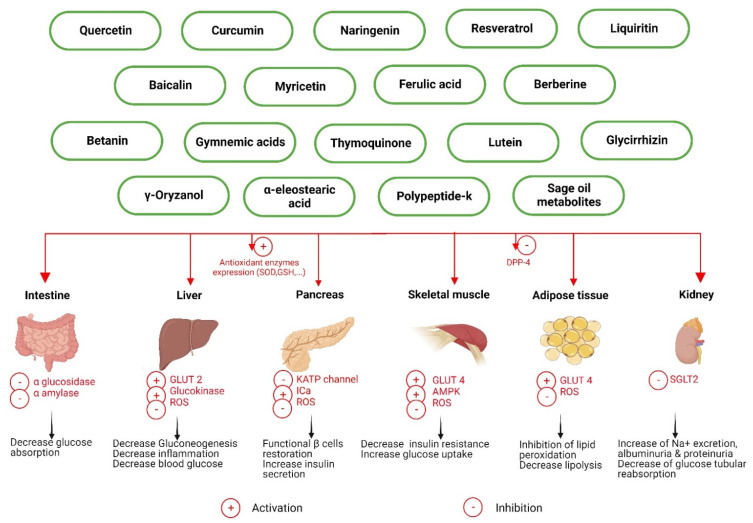
Various mechanisms and antidiabetic effects of some plant compounds (SOD—super oxide dismutase, GSH—glutathione, DPP-4—dipeptidyl peptidase 4, GLUT 2—glucose transporter 2, ROS—reactive oxygen species, KATP—adenosine triphosphate-sensitive potassium, ICa—calcium ion influx, GLUT 4—glucose transporter 4, AMPK—adenosine monophosphate-activated protein kinase, SGLT2—sodium-glucose cotransporter-2). Created with the use of BioRender.com.

**Figure 2 pharmaceutics-14-02135-f002:**
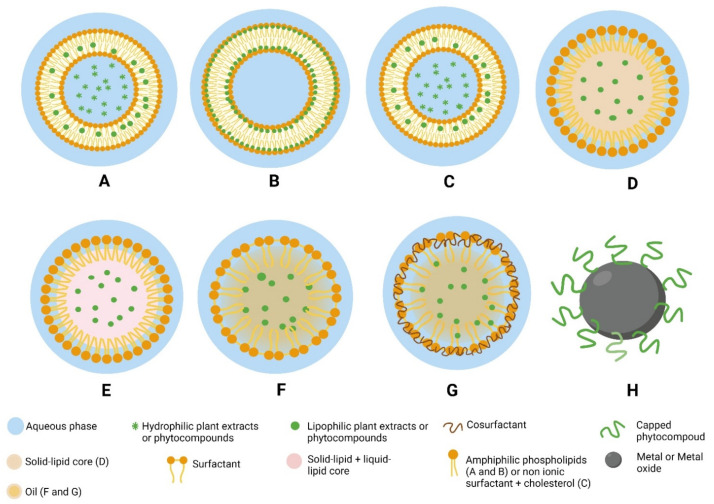
Structures of current lipid- and inorganic-based nanoparticulate delivery systems tested for antidiabetic herbal medicines. (**A**) Liposomes, (**B**) phytosomes, (**C**) niosomes, (**D**) solid lipid nanoparticles, (**E**) nanostructured lipid carriers, (**F**) nanoemulsions, (**G**) self-nanoemulsifying drug delivery systems, and (**H**) green-synthesized metallic or metal oxide nanoparticles. Created with the use of BioRender.com.

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
