# Peer review of "An Overview of Herbal-Based Antidiabetic Drug Delivery Systems: Focus on Lipid- and Inorganic-Based Nanoformulations"

_pharmaceutics, 2022, doi:10.3390/pharmaceutics14102135_

Round 1
Reviewer 1 Report
The manuscript presented a synopsis of herbal-based antidiabetic drug delivery systems with a precise focus on lipid-based and inorganic-based nanoformulations. A few concerns need to be addressed before assessing the suitability of the manuscript for publication. My specific comments are appended below.
1. The authors are appreciated for the extensive documentation, but the focus of the paper is diluted with redundant portions. More clarity on the emphasis of the manuscript is encouraged.
2. The safety of the enumerated traditional antidiabetic agents and specifically of the nanoformulations need to be substantially discussed.
3. The limitations of the traditional antidiabetic agents from the perspective of clinical use should be further elaborated.
4. The ‘Discussion & Conclusion’ part requires in-depth revision to make the review more systematic and comprehensive.
Author Response
The manuscript presented a synopsis of herbal-based antidiabetic drug delivery systems with a precise focus on lipid-based and inorganic-based nanoformulations. A few concerns need to be addressed before assessing the suitability of the manuscript for publication. My specific comments are appended below.
- The authors are appreciated for the extensive documentation, but the focus of the paper is diluted with redundant portions. More clarity on the emphasis of the manuscript is encouraged.
The manuscript has been cleaned up to remove repeating segments and to emphasize its importance.
- The safety of the enumerated traditional antidiabetic agents and specifically of the nanoformulations need to be substantially discussed.
We would like to thank the reviewer for this suggestion. Information regarding the safety of the enumerated phytocompounds has been added for each of them. They have been also commented in the discussion and conclusion.
- The limitations of the traditional antidiabetic agents from the perspective of clinical use should be further elaborated.
We agree with the reviewer’s comment. For each antidiabetic phytocompound the limitations are explained in its section. The main limitation is undoubtedly the low bioavailability. We have discussed this in the section intitled “Limitations of current antidiabetic phytocompounds”.
- The ‘Discussion & Conclusion’ part requires in-depth revision to make the review more systematic and comprehensive.
We have rewritten the "Discussion and Conclusion" section and improved it to make this review more comprehensive.
Reviewer 2 Report
The present review manuscript titled “An overview of herbal-based antidiabetic drug delivery systems: focus on lipid- and inorganic-based nanoformulations” by Kambale et al. is very well written. The authors cover almost all the aspects as per the objective of the review. Overall, the quality of the manuscript is excellent. However, the manuscript needs some revision. My comments are as follows.
Comment 1. The authors should discuss diabetes briefly in a separate section.
Comment 2. The authors should explain the molecular mechanism followed by phytochemicals in the anti-diabetic activity. Further, a good diagram should be drawn to illustrate the mechanism and easy understanding for the readers.
Comment 3. Figure 2 is not as per the journal standard. The structure of all the discussed nanocarriers should be included.
Author Response
The present review manuscript titled “An overview of herbal-based antidiabetic drug delivery systems: focus on lipid- and inorganic-based nanoformulations” by Kambale et al. is very well written. The authors cover almost all the aspects as per the objective of the review. Overall, the quality of the manuscript is excellent. However, the manuscript needs some revision. My comments are as follows.
Comment 1. The authors should discuss diabetes briefly in a separate section.
A section on diabetes has been created and the introduction slightly modified keeping our aim.
Comment 2. The authors should explain the molecular mechanism followed by phytochemicals in the anti-diabetic activity. Further, a good diagram should be drawn to illustrate the mechanism and easy understanding for the readers.
We agree with the reviewer’s comment. The graph has been newly drawn and the antidiabetic mechanisms of phytochemicals have been added. In each phytocompound section the mechanisms are briefly given.
Comment 3. Figure 2 is not as per the journal standard. The structure of all the discussed nanocarriers should be included.
We thank the reviewer for this comment. Figure 2 has been redone according to the journal's standards and all discussed nanocarriers have been included within the figure.
Round 2
Reviewer 1 Report
The authors have revised the manuscript substantially. The current version is now recommended for publication.
Reviewer 2 Report
I appreciate the authors for addressing all the comments very carefully. I don't have further comments.